# New Approach to the Cryopreservation of GV Oocytes and Cumulus Cells through the Lens of Preserving the Intercellular Gap Junctions Based on the Bovine Model

**DOI:** 10.3390/ijms25116074

**Published:** 2024-05-31

**Authors:** Taisiia Yurchuk, Pawel Likszo, Krzysztof Witek, Maryna Petrushko, Dariusz J. Skarzynski

**Affiliations:** 1Department of Reproductive Immunology and Pathology, Institute of Animal Reproduction and Food Research of Polish Academy of Sciences in Olsztyn, 10-748 Olsztyn, Poland; p.likszo@pan.olsztyn.pl (P.L.);; 2Department of Cryobiology of Reproductive System, Institute for Problems of Cryobiology and Cryomedicine of the National Academy of Sciences of Ukraine, 61-016 Kharkiv, Ukraine

**Keywords:** cumulus cell cryopreservation, GV oocyte in vitro maturation, vitrification, Cx37, Cx43, gap junction, cell communication, apoptosis, epigenetic changes, female fertility preservation

## Abstract

Differences in structural and functional properties between oocytes and cumulus cells (CCs) may cause low vitrification efficiency for cumulus–oocyte complexes (COCs). We have suggested that the disconnection of CCs and oocytes in order to further cryopreservation in various ways will positively affect the viability after thawing, while further co-culture in vitro will contribute to the restoration of lost intercellular gap junctions. This study aimed to determine the optimal method of cryopreservation of the suspension of CCs to mature GV oocytes in vitro and to determine the level of mRNA expression of the genes (*GJA1*, *GJA4*; *BCL2*, *BAX*) and gene-specific epigenetic marks (*DNMT3A*) after cryopreservation and in vitro maturation (IVM) in various culture systems. We have shown that the slow freezing of CCs in microstraws preserved the largest number of viable cells with intact DNA compared with the methods of vitrification and slow freezing in microdroplets. Cryopreservation caused the upregulation of the genes Cx37 and Cx43 in the oocytes to restore gap junctions between cells. In conclusion, the presence of CCs in the co-culture system during IVM of oocytes played an important role in the regulation of the expression of the intercellular proteins Cx37 and Cx43, apoptotic changes, and oocyte methylation. Slow freezing in microstraws was considered to be an optimal method for cryopreservation of CCs.

## 1. Introduction

Cryopreservation of reproductive cells, tissues, and embryos has become an integral part of assisted reproductive technologies (ART), mainly for preservation of the fertility of patients [1]. The success of using cryobiological methods is determined by the extent to which the biological object retains its properties after all stages of cryopreservation: equilibration/removal of the cryoprotectant solutions, cooling, and thawing. Maintenance of the morphological and functional features of the gametes, embryo, and accessory cells after cryopreservation and thawing is a key factor affecting the success of ART. Thus, it has been proven that vitrification is a more effective method of preserving oocytes and embryos compared with slow freezing, and a fast two-stage method is more effective for spermatozoa [2,3,4,5]. Slow freezing is considered a standard method for cryopreservation of tissues and suspensions of somatic cells [6]. In the case of tissues, the effectiveness of the freezing method can be lower, since they are composed of different types of cells that differ in their morphology and function, but also in their cryosensitivity. The standard method for the cryopreservation of ovarian tissue, with oocytes of different degrees of maturity, is considered to be slow freezing [7]. However, in the case of freezing tissue, some problems are associated with the occurrence of ischemia and the restoration of vascularization after transplantation [8]. Given this, cryopreservation of individual follicles or immature oocytes with subsequent in vitro maturation (IVM) has advantages but requires deeper study and improvement [9,10,11].

It is well known that oocytes in the process of maturation are surrounded by a few layers of CCs, which perform the function of transporting nutrients and signaling molecules, including through the gap junctions, which play an extremely important role in the maturation of oocytes [12,13,14,15,16]. The gap junctions between oocytes and CCs are formed mainly by transmembrane proteins (connexin 37 (Cx37)) and between CCs (connexin 43 (Cx43)) [17]. Therefore, during cryopreservation, especially of immature oocytes, it is necessary to ensure the preservation of communication between cells. High concentrations of penetrating and non-penetrating cryoprotectants and ultra-high cooling and thawing speeds are used for the vitrification of oocytes to reach the glass state condition and minimize damage to the gametes. At the same time, the presence of several dense layers of CCs around an immature oocyte, on the one hand, slows down its equilibration with a cryoprotective solution; therefore, it may take a longer time [18]. On the other hand, CCs, compared with oocytes, are characterized by much smaller size and a low ratio of surface area to cell volume, resulting in much faster equilibration with the cryoprotective solution, and further exposure can have a cytotoxic effect on CCs [18,19,20]. Osmotic changes in the cells during vitrification of COCs and damage lead to disturbances of the intercellular contacts, which can cause low viability and metabolic changes not only in the CCs but also in the oocyte, and compromise the resumption of meiosis during IVM [21]. Moreover, vitrification causes significant increases in DNA fragmentation in CCs, which may affect further maturation of the oocytes [22]. Nowadays, cryopreservation of CCs is considered in the context of the preservation of oocytes. In most studies where vitrification has been used for the cryopreservation of COCs, a low survival rate, viability, and functionality of CCs after thawing was shown [23]. The structural and functional differences in oocytes and CCs require different approaches to cryopreservation. To do this, it is necessary to separate the CCs from the oocytes before the cryopreservation procedure and create conditions for the restoration of intercellular communication after thawing.

This study aimed to determine the optimal method of cryopreservation of the suspension of CCs to mature GV oocytes in vitro and to determine the level of expression of the mRNA of genes (*GJA1*, *GJA4*; *BCL2*, *BAX*) and gene-specific epigenetic marks (*DNMT3A*) after cryopreservation and IVM in various culture systems. In our work, we used a bovine model to evaluate all the abovementioned parameters before applying a new approach for preservation and IVM of immature human oocytes.

## 2. Results

### 2.1. Effect of the Cryopreservation Method on the Viability and DNA Integrity of CCs

At the first stage of our study, we compared the efficiency of using different methods (vitrification, slow freezing in microdrops, and slow freezing in microstraws) for preservation of CCs. The viability was measured in CCs by Trypan blue staining after warming. As shown in Figure 1A, the highest value of this parameter was obtained after slow freezing in microstraws. There was no significant difference between the study parameters after vitrification and slow freezing in microdrops.

On the contrary, the DNA fragmentation rate was the lowest after slow freezing in microstraws (Figure 1B) (*p* < 0.05) and did not differ significantly from freshly isolated cells. The highest level of DNA damage was registered in CCs after slow freezing in microdrops, while, after vitrification, it was significantly higher only compared with the control group, namely fresh CCs (fCCs) (*p* < 0.05).

According to the data obtained, the method of slow freezing in microstraws in cryovials was used in the subsequent research for cryopreservation of CCs.

### 2.2. Effect of Cryopreservation of Oocytes and CCs and IVM Conditions on Intercellular Connections and the Level of mRNA Expression of Cx37 and Cx43

Figure 2 shows photos of the cells before and after IVM. In the fCOCs group, a large-scale expansion of CCs was observed, while in the cryoCOCs group, morphological alterations in the COCs were observed immediately after thawing, in the form of compaction of CCs, their detachment from the oocyte, and their degradation (Figure 2E), which increased after IVM (Figure 2F). No morphological differences were observed between the fDOs + fCCs and cryoDOs + cryoCCs groups after the oocytes’ maturation (Figure 2G,H). CCs aggregated between themselves and oocytes, forming a structure similar to COCs (Figure 2E). It turned out that cryopreservation did not affect the ability of CCs to form aggregates, since similar properties were also manifested in the group where fCCs were used.

After the maturation of DOs (both fresh and cryopreserved), the expression rate of Cx37 mRNA was significantly higher compared with all other groups studied (*p* < 0.05) (Figure 3A). Oocytes from the fCOC and fDOs + fCCs groups were characterized by a higher level of the mRNA expression of Cx37 compared with immCOCs and cryoCOCs and cryoDOs + cryoCCs oocytes. The expression level of the Cx37 mRNA was significantly higher in CCs after maturation in all study groups except fCOCs, compared with immature CCs (*p* < 0.05). It should also be noted that the presence of oocytes in the culture system and/or cryopreservation did not affect the transcriptome level of this gene (Figure 3B).

Immunofluorescent staining of the samples showed that Cx37 is mostly localized in the cytoplasm of the oocyte, as well as in CCs (Figure 3C–S). The distribution of this protein changed only in the DOs group, where the fluorescence of the Cx37 antibody’s fluorochrome was slightly detected (Figure 3N,O). After cryopreservation of the DOs, the visualization of Cx43 was extremely weak (Figure 3R,S).

The study of the expression of Cx43 mRNA in the oocytes showed that its level did not increase after IVM compared with immCOCs (*p* > 0.05) (Figure 4A). However, in the fDOS and cryoDOs groups, the level of Cx43 transcripts was significantly higher (*p* < 0.05).

After cryopreservation, the level of Cx43 increased in all study groups (Figure 4B). Analysis of the preparations after immunofluorescence staining showed that Cx43 was localized at the border between the neighboring CCs and the zona pellucida (ZP); in addition, the signal was also detected in the ooplasm of immature cells and after their maturation (Figure 4C,D,F,G). In the fDOs + fCCs and fDOs groups, the signal was detected closer to the ZP (Figure 4K,O). After cryopreservation, the localization of Cx43 shifting to the ZP region occurred in the cryoCOCs and cryoDOs + cryoCCs groups. The transzonal localization of Cx43 was especially noted in the group of cryoDOs (Figure 4Q).

### 2.3. Effect of Cryopreservation of Oocytes and CCs and IVM Conditions on mRNA Expression Levels of BCL2, BAX, and DNMT3A

The expression levels of *BAX* mRNA in fresh oocytes increased significantly after maturation in fCOCs, fDOs, and cryoDOs groups (Figure 5A). The level of *BCL2* was the highest in the fCOCs group and was significantly different from all other groups (*p* < 0.05). Therefore, the *BAX/BCL2* ratio in the fCOCs group was the lowest. The highest expression level of *BAX* was found in the fDOs group, which increased after cryopreservation. Considering the low expression of *BCL2* in the cryoDOs group, the *BAX/BCL2* ratio was also very high.

In the CCs from immCOCs, as in oocytes, a high *BAX/BCL2* ratio was noted due to the very low expression of *BCL2* (Figure 5F). The highest level of *BAX* transcriptomes was determined in CCs after maturation of fCOCs and cryoDOs + cryoCCs; however, the *BAX/BCL2* ratio was low only in the fCOCs group due to the high expression of *BCL2*.

### 2.4. Effect of Cryopreservation of Oocytes and CCs and IVM Conditions on the mRNA Expression of DNMT3A

The highest level of mRNA expression of *DNMT3a* was found in DOs after IVM, and cryopreservation did not significantly change this indicator (Figure 6A). After maturation, the *DNMT3a* transcriptome’s level did not change in fCOCs compared with immCOCs. However, the mRNA expression of this gene was significantly different in the fCOCs and cryoCOCs groups. *DNMT3a* mRNA levels increased significantly after IVM in all groups where CCs were used, regardless of cryopreservation (*p* < 0.05) (Figure 6B).

### 2.5. Effect of Cryopreservation of Oocytes and CCs and Conditions on the Outcomes of IVM

The highest oocyte maturation rate was observed in the fCOCs group; at the same time, cryopreservation led to a significant decrease in this indicator (Figure 7). The maturation rate of fDOs in the system of co-culture with fCCs resulted in 62% gametes at the metaphase II stage. However, although this indicator decreased to 39% when cryopreserved cells were used in the same culture system, it was significantly higher than in the cryoCOCs group, which was 19%. The lowest maturation rates were in groups of DOs, both fresh and after cryopreservation.

## 3. Discussion

In our study, we compared the efficiency of cryopreservation of CCs after vitrification by the standard method and slow freezing using different cryocarriers. It turned out that the choice of cryocarrier significantly affected the effectiveness of their cryopreservation. Thus, cryopreservation of CCs in microdroplets resulted in significant damage, which proved to be lethal, compared with when microstraws were used. At the same time, vitrification also negatively affected CCs, since the level of viability of CCs did not differ from the group with slow freezing in microdrops.

Since the DNA fragmentation test of CCs is used as a test of oocytes’ competence and, given that cryopreserved CCs were considered a component of the co-culture system for oocytes’ IVM, we evaluated this parameter in cells after thawing [24]. The most dramatic damage to DNA was detected in CCs when slow cooling was applied in microdroplets. The use of microstraws made it possible to preserve a significantly bigger number of cells with intact DNA, which did not exceed this indicator in fresh cells. Vitrified CCs had an increased level of DNA fragmentation, as shown in equine [25], porcine [26], and feline [27] CCs.

The cause of cell damage due to vitrification is most likely to be high concentrations of cryoprotectant agent, while during slow freezing, the damage is caused mostly by intracellular ice crystals [28]. The thawing stage is one of the most important in cryopreservation, as there is a risk of recrystallization [29]. Perhaps, thawing microstraws directly in a large volume of preheated solution makes it possible to develop the optimal speed of thawing of the sample to minimize the processes of recrystallization and subsequent damage. Ultimately, this helps preserve more viable cells with intact DNA at the level of freshly isolated cells.

In our next experiment, we evaluated how cryopreservation affected the intercellular contacts and their restoration during IVM, the level of expression of pro- and antiapoptotic genes, and methylation markers. As a result of our research, we found that vitrification disrupted the structure of COCs and changed the dynamics of CCs’ expansion. The addition of both cryoCCs and fCCs to the oocyte maturation system resulted in the formation of cell conglomerates not only between CCs but also between CCs and the oocytes. Moreover, the ability of CCs to create conglomerates was not affected by the presence of oocytes in the co-culture system.

The experiment on the effect of cryopreservation and different culture systems on the expression level of *GJA4* (Cx37) mRNA showed that its maximum level was in cryopreserved oocytes without CCs. Presumably, as a result of the breakdown of connections with CCs and the deficiency in regulatory signals, the expression level of the *GJA4* gene became maximal. Weak visualization of Cx37 in this group confirmed the possibility of a compensatory reaction of cells due to the loss of these connections. It should also be noted that a higher level of expression of the specified gene in the oocytes was seen in the group of fCOCs and fDO + fCCs compared with immCOCs. In this case, most likely, another mechanism was triggered, involving interactions between CCs and the oocyte. The expression of Cx37 was increased in CCs after culturing in IVM medium in all study groups except fCOCs. We assumed that partial or complete disconnection of oocytes and CCs occurred as a result of denudation or cryopreservation, which could lead to compensatory upregulation of the expression of Cx37 in the cells.

The expression of *GJA1* (Cx43) revealed a high level of mRNA in DOs, both fresh and cryopreserved, and its localization was mostly in the transzonal region, which was not observed in the groups of immCOCs and fCOCs, where Cx43 was located mostly in the ooplasm. It can be assumed that, in this way, oocytes try to establish lost contacts with CCs and actively build channels for communication with them. The expression of the Cx43 gene in CCs decreased after maturation in groups without cryopreservation compared with immCOCs, which is physiologically consistent from the point of view of the completion of nuclear maturation. This is consistent with the results of other researchers who showed the same effect on human CCs [30]. Moreover, lower levels of Cx43 mRNA in CCs have been shown to correlate with better embryological outcomes after fertilization of oocytes [31]. Increased levels of Cx43 mRNA were observed in cryopreserved COCs in our study, most likely as a compensatory response to the disconnection of oocytes and CCs. The absence of a significant difference in the co-culture of cryopreserved oocytes and CCs indicated the possibility of reconstruction of the contact between cells, which can positively affect the oocytes’ further IVM.

To summarize our data on the expression of gap junction proteins, the level of expression of the Cx37 gene in the oocytes depended on the presence of CCs in the IVM culture. Thus, in the DOs group, where CCs were not added, the level of mRNA expression of this protein increased, probably in a compensatory manner. Cryopreservation reduced the level of mRNA expression of Cx37 in oocytes in the presence of CCs in the IVM culture system to the level of immature COCs. While in CCs, an increased level of Cx37 expression was observed after cultivation in the IVM medium, except for the group of CCs where the they were part of fCOCs. Similar to Cx37, the expression level of Cx43 mRNA in oocytes during maturation was influenced by the presence of CCs in the IVM culture system. Denuded oocytes tried to restore the lost intercellular connections and changed the localization of Cx43 from the ooplasmic to the transzonal region. This effect was much more pronounced in denuded oocytes after cryopreservation.

It is well known that the *BAX/BCL2* ratio determines the level of cell apoptosis. A high ratio of *BAX/BCL2* will stimulate the effector and executive stages of apoptosis via the intrinsic pathway [32]. In our study, a specific increase in the antiapoptotic gene *BCL2* was observed in oocytes matured in fCOCs. Other authors showed that the expression of BCL2 correlates with oocytes of high quality [33]. We showed that after cryopreservation and denudation of the oocytes, the ratio of *BAX/BCL2* increased towards the development of apoptosis, and the presence of CCs in the co-culture system or as a component of COCs affected the decrease in this parameter. However, the activity of *BCL2* was significantly lower in these groups.

*DNMT3A* plays an important role in establishing and maintaining DNA methylation during oocytes’ maturation and affects the expression of many genes [34]. We evaluated its mRNA level after exposure to cryopreservation and different IVM systems. The low level of DNMT3A in immCOCs that we obtained is consistent with the results described previously, which demonstrated a general decrease in transcription at the end of the oocytes’ growth phase and during the transition to meiotic competence [35]. This is evolutionarily possible due to the accumulation of stable mRNA at the beginning of oocytes’ development, which becomes a reservoir for their further use to support the oocytes’ maturation and embryogenesis. Silencing of transcriptional activity is believed to begin in the GV phase and is not fully reversed until fertilization [35]. Our work showed that the mRNA level of *DNMT3A* was significantly higher in DOs regardless of the effect of cryopreservation. Thus, the presence of CCs in the co-cultivation system or as a component of COCs most likely could keep methylation levels in the oocytes at the basic stage. At the same time, the level of *DNMT3A* increased in CCs during culturing in the IVM medium regardless of the presence of oocytes, which may indicate a change in the CCs’ gene activity caused by hormones and growth factors of the IVM medium. Cryopreservation did not lead to changes in the expression of *DNMT3A* in CCs. However, cryopreservation of COCs led to a decrease in epigenetic markers in the oocytes compared with their fresh counterparts.

The maturation rate of oocytes was maximal in fCOCs, while cryopreservation led to a detrimental drop in this indicator (by almost fourfold). The lowest maturation rate was observed in denuded oocytes, both fresh and cryopreserved, which can be explained by the role of CCs in the maturation processes of oocytes [36]. Co-culture of oocytes with CCs had a positive effect on the oocytes’ maturation. Adding cryopreserved CCs to the co-culture system reduced this indicator (*p* < 0.05), but it was higher than in the cryoCOCs and cryoDOs groups. Thus, we may conclude that the presence of viable CCs during maturation is extremely important for the resumption of meiosis in oocytes.

## 4. Materials and Methods

### 4.1. Retrieval of Cumulus–Oocyte Complexes

Ovaries were derived from mixed-breed cows and transported from a local slaughterhouse in a warm saline solution (35 °C) to a laboratory; COCs were retrieved from follicles ranging in diameter from 2 to 8 mm using an 18 gauge needle attached to a 10 mL syringe. Only COCs with at least 3 layers of CCs were selected and used in the experiments.

### 4.2. Experimental Design

To determine the optimal method of cryopreservation of the suspension of CCs, the obtained cells were divided into the following groups: 1, vitrified; 2, frozen in microdrops by a slow method; 3, frozen in microstraws by a slow method (Figure 8). CCs’ viability and levels of DNA fragmentation were determined after warming.

To determine the effect of cryopreservation and the method of IVM on the mRNA level, the expression of genes that are important for the oocytes’ development and maturation, and the localization of Cx37 and Cx43, all retrieved COCs were randomly divided into the following groups and subjected to the appropriate procedures: fresh immature COCs (immCOCs), fresh COCs after IVM (fCOCs), cryopreserved COCs after IVM (cryoCOCs), freshly denuded oocytes after IVM (fDOs), cryopreserved denuded oocytes after IVM (cryoDOs), fresh denuded oocyte matured in vitro with a fresh suspension of CCs (fDOs + fCCs), cryopreserved oocytes matured in vitro with a cryopreserved suspension of CCs (cryoDOs + cryoCCs), fresh CCs after culturing in IVM media (fCCs), cryopreserved cumulus cells after culturing in IVM media (cryoCCs) (Figure 9).

### 4.3. Denudation of Oocytes

COCs were immersed in a solution of hyaluronidase (FertiPro, Beernem, Belgium) for 1 min, after which, they were washed 2 times in a wash solution environment, which consisted of a medium with 199 HEPES (Gibco, Grand Island, NY, USA) and 5% FBS (Gibco, Grand Island, NY, USA). Part of the collected COCs was denuded with a stripper. Then the oocytes underwent IVM or cryopreservation.

### 4.4. Cryopreservation of COCs and Oocytes

COCs or DOs were vitrified using the Cryotec method. For this, the cells were immersed in a solution containing 7.5% DMSO, 7.5% ethylene glycol, and 20% FBS based on the 199 HEPES medium for 7 min, after which they were transferred to a vitrification solution containing 15% DMSO, 15% EH, 0.6 M, sucrose and 20% FBS. After 1 min, the cells were placed on the carrier of the Cryotec (Cryotech, Tokyo, Japan) and immersed in liquid nitrogen, where they were stored for at least 1 week. Thawing was carried out in a solution of 0.6 M sucrose based on the medium of 199 HEPES heated to a temperature of 38.5 °C over 1 min, then the samples were transferred to a washing solution and the IVM medium.

### 4.5. CCs Suspension’ Cryopreservation Procedure and Their Viability Assessment

The extracted CCs after denudation of the oocytes were collected in Eppendorf tubes. After centrifugation, the supernatant was discarded and Accumax (Capricorn Scientific, Ebsdorfergrund, Germany) was added to the pellet, gently mixed, and incubated over 3 min at 38.5 °C. Then the CCs were washed twice with the wash solution. The cells were evaluated for viability with the help of trypan blue. The freshly obtained suspension of CCs was added to DOs or cryopreserved. Cryopreservation of the CCs was carried out by three different methods: slow freezing in drops of 10 μL in cryovials (Nunc, Roskilde, Denmark); the same volume collected in microstraws (Figure 10), which were located in cryovials; or vitrification. The cryoprotectant media for CCs consisted of 10% DMSO, and 20% FBS in the 199 medium with l-L-glutamine and HEPES. As a microstraw, we used disposable capillary pipettes manufactured for manipulation of oocytes and embryos (Minitube, Germany). They were cut to fit inside 1.8 mL cryovials (Nunc, Denmark) (Figure 10). The CCs were equilibrated with the cryoprotectant solution in microstraws at room temperature for 5 min and placed in the cryovials. They were cooled down in a CoolCell^®^ container (Corning, Corning, NY, USA) in a freezer (−80 °C). After 24 h, all samples were transferred to liquid nitrogen for further storage. The vitrification of CCs was performed as described in the previous section. Thawing of the samples which were frozen slowly in microdrops was carried out by immersing the vial in a water bath at a temperature of 38.5 °C; after that, the cells were twice washed in a wash solution. Vials with microstraws were removed from the liquid nitrogen and, after 30 s, the straw was immersed in the wash solution (38.5 °C) and washed twice. The viability of cryopreserved cells was evaluated after warming, which was expressed as a percentage of the initial viability of the cell suspension. The calculations were carried out at least five times for each study group.

### 4.6. DNA Fragmentation of CCs

The level of DNA fragmentation in the CCs was assessed by the TUNEL test using an In Situ Cell Death Detection Kit (Roche, Brussel, Belgium) according to the manufacturer’s instructions. The positive and negative controls were incubated with DNase I (deoxyribonuclease I, 100 IU/mL; Invitrogen, Carlsbad, CA, USA) at 37 °C for 1 h. Then, after washing in PBST, cells from the positive control were incubated with the TUNEL reaction mixture containing FITC-conjugated dUTP (label solution) and terminal deoxynucleotidyl transferase at 37 °C for 1 h in the dark. The negative control was incubated only with the label solution in the absence of terminal deoxynucleotidyl transferase. All samples were stained with 10 mg/mL Hoechst (33342, Sigma–Aldrich, St. Louis, MO, USA) in PBS for 15 min at room temperature. The excitation wavelength of 488 nm was used to identify TUNEL staining and the 352 nm wavelength was used to distinguish the nuclei (Figure 1). In each sample, at least 200 cells were assessed by TUNEL. The experiment was repeated three times.

### 4.7. Maturation of COCs and Oocyte–CCs Co-Culture System

The obtained COCs (*n* = 1019) were divided into groups and matured in vitro in a drop of IVM medium (Stroebech, Copenhagen, Denmark) with a volume of 100 μL under mineral oil. The total number of COCs subjected to the IVM was 187 and 124 for the fCOCs and cryoCOCs groups, respectively. DOs were cultured without CCs (fDOs, *n* = 194; cryoDOs, *n* = 153). For co-culture, a suspension of CCs at a concentration of 5 × 10^5^/mL was added to the oocytes in a drop of the same volume (oocytes, *n* = 216 for the fDOs + fCCs group; oocytes, *n* = 145 for the cryoDOs + cryoCCs group). The maturation rate was estimated after 22 h by calculating the percentage of oocytes with an extrusion of the first polar body. The measurements of the maturation rate were repeated three times.

### 4.8. Immunofluorescent Staining

COCs, DOs, and CCs were fixed in 4% paraformaldehyde (Biotium, Fremont, CA, USA) for 30 min, after which, they were washed in PBS solution with 0.05% of Tween 20 (Sigma-Aldrich, St. Louis, MO, USA). Permeabilization was performed with a solution of 0.05% Triton X-100 over 1 h and blocked with fish serum blocking buffer (ThermoFisher Scientific, Waltham, MA, USA). As primary antibodies, we used Cx37 antibody Cat. No. #42-4400 (ThermoFisher Scientific, Waltham, MA, USA) at a 1:100 ratio. After washing twice in the fish serum blocking buffer, the cells were incubated for 1 h with the secondary antibodies goat anti-rabbit IgG (H + L) and Alexa Fluor™ 594 and 488 (ThermoFisher Scientific, Waltham, MA, USA) in a 1:200 ratio. Cells were counterstained with DAPI (ThermoFisher Scientific, Waltham, MA, USA) at a concentration of 10 μg/mL for 15 min.

### 4.9. RNA Isolation, cDNA Synthesis, and Real-Time qPCR

COCs or DOs and CCs were subjected to RNA isolation using the PicoPure RNA Isolation Kit (#KIT0204, Applied Biosystems, Waltham, MA, USA) according to the manufacturer’s instructions. The RNA samples were stored at −80 °C. The RNA content in each sample was 60 ng. The resulting complementary DNA (cDNA) was stored at −20 °C for further quantitative PCR (qPCR) analysis. TaqMan Universal Master Mix II (Life Technologies) was used. For each of the genes (Table 1), the PCR mix consisted of 3 μL of cDNA (equal to 2 ng), 0.5 μL of the TaqMan assay, 5 μL of the TaqMan master mix, and 1.5 μL of RNase-free water. The mRNA transcript levels were normalized to *GAPDH* (internal control). The 384-well plates were prepared using the Bravo automated liquid handling platform (Agilent Technologies, Inc., Santa Clara, CA, USA). The analysis was performed in duplicate for each sample, in a final volume of 10 μL, using the ABI Prism 7900 HT Real-Time PCR System. The thermal profile used to amplify the genes was as follows: polymerase activation (95 °C for 10 min), followed by 45 cycles of denaturation (95 °C for 15 s) and annealing (60 °C for 1 min). Real-time PCR Miner Software 4.0 (118.190.66.83) [18] was used to estimate the mean efficiency of PCR amplification and the cycle threshold (Ct) values for each gene. We conducted three biological replicates for the expression of each gene in each group.

### 4.10. Statistical Data Processing

Statistical analysis was conducted using GraphPad Prism Software (Version 9.02, GraphPad Software Inc., La Jolla, CA, USA). In all experiments, the samples were tested for normality via Shapiro–Wilk tests. In all the experiments, the samples were analyzed by one-way ANOVA with a post hoc Tukey’s test. The differences were considered statistically significant at *p* < 0.05.

## 5. Conclusions

Slow freezing of CCs in microstraws made it possible to preserve the largest number of viable cells with intact DNA, which can reproduce the contacts between oocytes and other CCs, which allowed this method to be considered the optimal.

The obtained data from the study gave a new idea of cell communication during oocytes’ IVM. Co-culture of cryopreserved oocytes and CCs allowed us to obtain a higher rate of maturation compared with cryopreserved COCs or DOs which were cultured without the addition of CCs. The proposed model of IVM using separately cryopreserved oocytes and CCs requires further improvement to effectively restore lost gap junctions and communication between CCs, which contribute to the development of competent oocytes. The reconstruction of COCs from cryopreserved CCs and oocytes will provide new opportunities for preservation of the fertility of patients who have contraindications for the use of superovulation induction or have certain oocyte IVM dysfunctions.

## Figures and Tables

**Figure 1 ijms-25-06074-f001:**
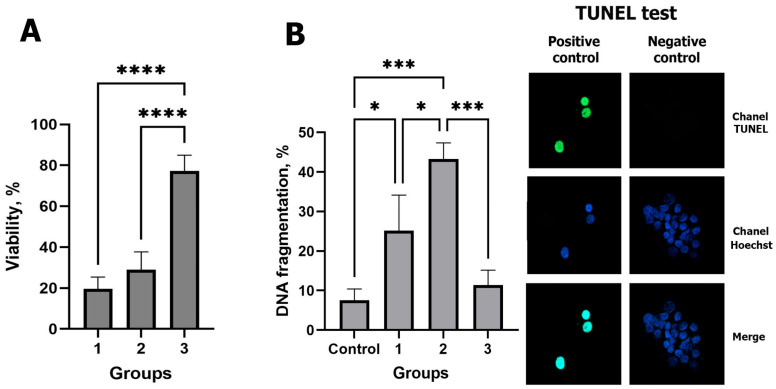
The effect of cryopreservation on CCs’ viability and DNA fragmentation. (**A**) Viability and (**B**) DNA fragmentation rate determined by the TUNEL test. Group 1, vitrification; Group 2, slow freezing in microdrops; Group 3, slow freezing in microstraws, control, suspension of fCCs. The asterisks mean *p* value: *, ≤0.05; ***, ≤0.001; ****, ≤0.0001.

**Figure 2 ijms-25-06074-f002:**
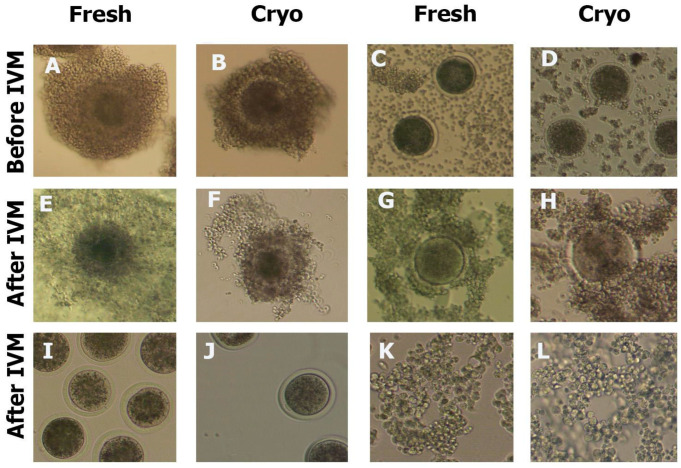
Morphology of oocytes and CCs of cows cryopreserved and cultured in various in vitro systems. (**A**) ImmCOCs immediately after retrieval; (**B**) immCOCs immediately after warming before IVM; (**C**) fresh immDOs with CCs from immCOCs before IVM; (**D**) cryopreserved immDOs with CCs from immCOCs before IVM; (**E**) fCOCs after IVM; (**F**) cryoCOCs after IVM; (**G**) fDOs with CCs after IVM; (**H**) cryoDOs with cryoCCs after IVM; (**I**) fDOs after IVM; (**J**) cryoDOs after IVM; (**K**) fCCs cultured for 24 h in IVM medium; (**L**) cryoCCs cultured for 24 h in IVM medium. Descriptions of the groups can be found in Section 4.2 (Experimental Design).

**Figure 3 ijms-25-06074-f003:**
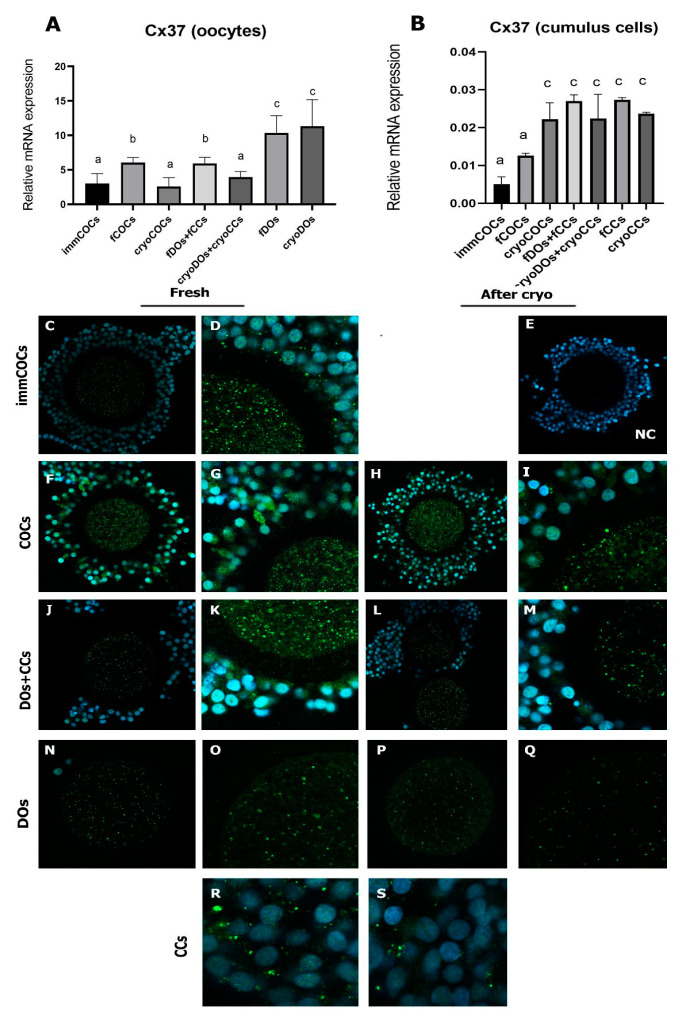
Expression of Cx37 in oocytes and CCs after cryopreservation and in vitro maturation in different co-culture systems. (**A**) Relative mRNA expression of Cx37 in oocytes; (**B**) relative mRNA expression of Cx37 in CCs; (**C**–**S**) immunofluorescent staining of Cx37 (green) and nucleus counterstaining with DAPI (blue). (**C**,**D**) ImmCOCs; (**E**) negative control; (**F**,**G**) COCs; (**H**,**I**) cryoCOCs; (**R**) fCCs; (**S**) cryoCCs. Descriptions of the groups can be found in Section 4.2 (Experimental Design). Different subscript letters on the graph mean statistically significant difference between the groups (*p* ≤ 0.05).

**Figure 4 ijms-25-06074-f004:**
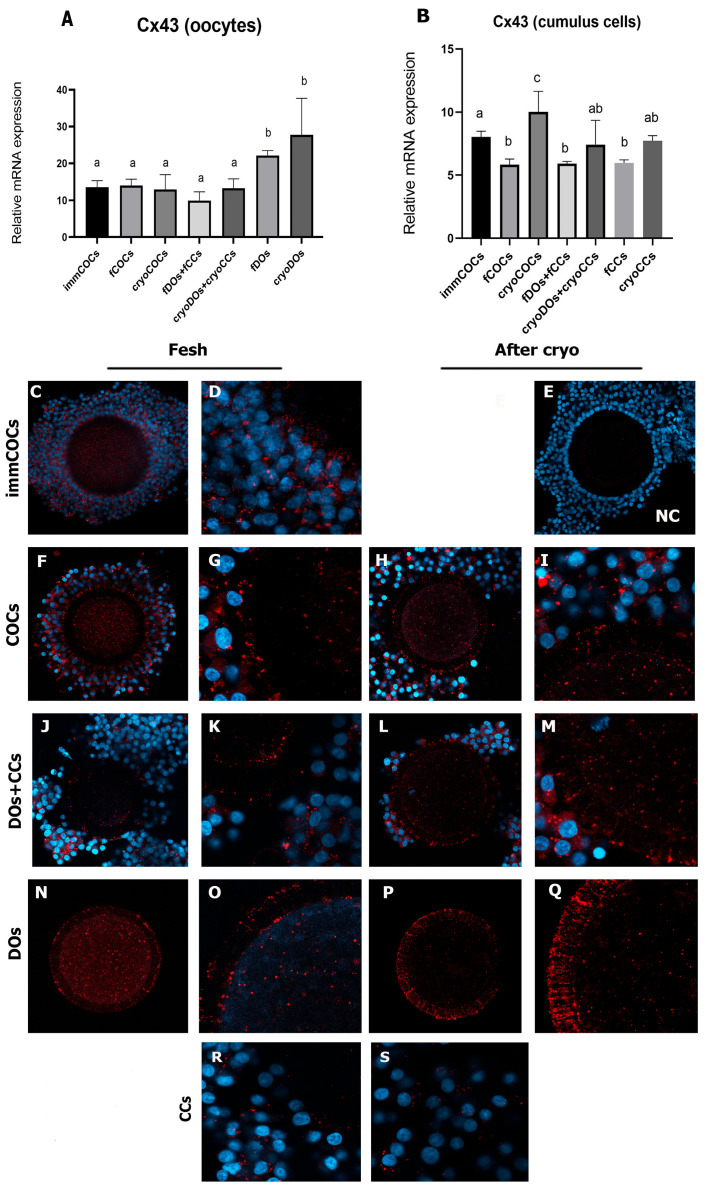
Expression of Cx43 in oocytes and CCs after cryopreservation and in vitro maturation in different co-culture systems. (**A**) Relative mRNA expression of Cx43 in oocytes; (**B**) relative mRNA expression of Cx43 in CCs; (**C**–**S**) immunofluorescent staining of Cx43 (red) and counterstaining of the nucleus with DAPI (blue). (**C**,**D**) ImmCOCs, (**E**) negative control, (**F**,**G**) COCs, (**H**,**I**) cryoCOCs, (**R**) fCCs, (**S**) cryoCCs. Descriptions of the groups can be found in Section 4.2 (Experimental Design). Different subscript letters on the graph mean statistically significant difference between the groups (*p* ≤ 0.05).

**Figure 5 ijms-25-06074-f005:**
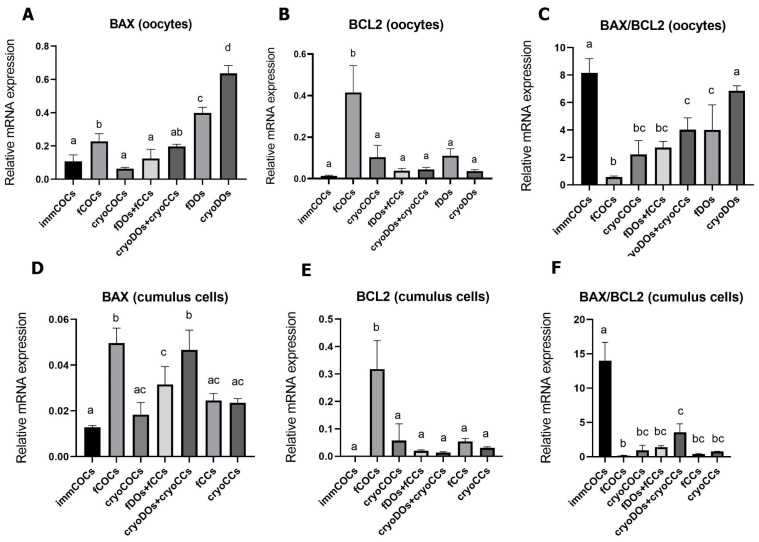
Expression of *BAX* and *BCL2* genes and the *BAX/BCL2* ratio in oocytes (**A**–**C**) and CCs (**D**–**F**) after cryopreservation and in vitro maturation in different co-culture systems. Descriptions of the groups can be found in Section 4.2 (Experimental Design). Different subscript letters on the graph mean statistically significant difference between the groups (*p* ≤ 0.05).

**Figure 6 ijms-25-06074-f006:**
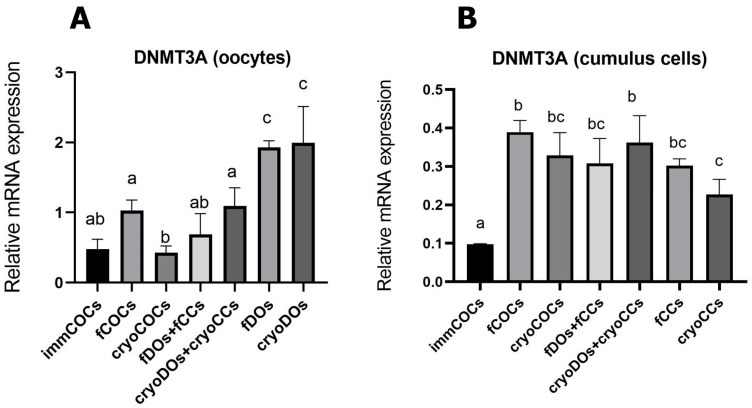
Expression of the *DNMT3A* gene in oocytes (**A**) and CCs (**B**) after cryopreservation and in vitro maturation in different co-culture systems. A description of the groups can be found in Section 4.2 (Experimental Design). Different subscript letters on the graph mean statistically significant difference between the groups (*p* ≤ 0.05).

**Figure 7 ijms-25-06074-f007:**
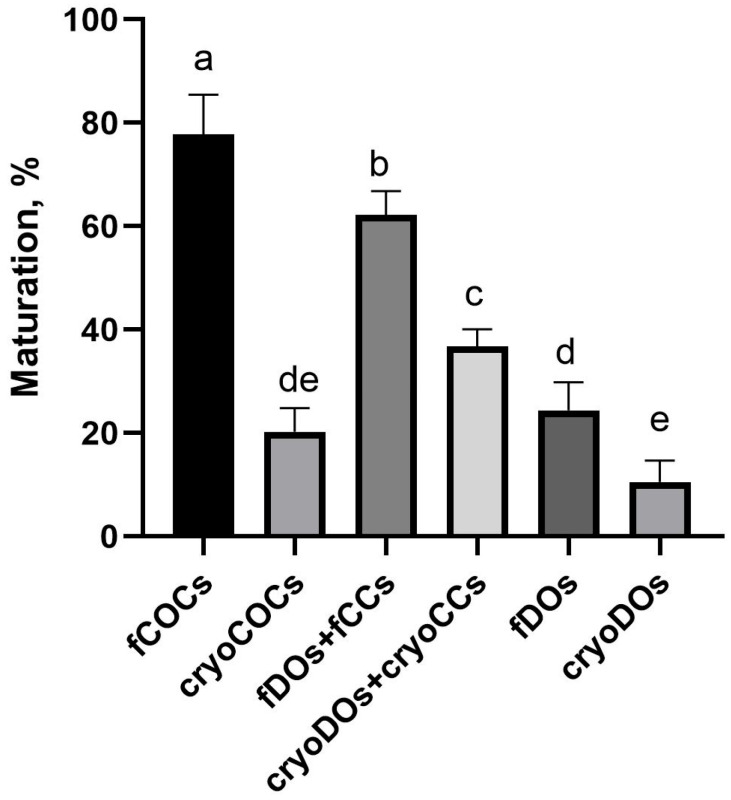
Maturation rate of oocytes after cryopreservation and in vitro maturation in different co-culture systems. Descriptions of the groups can could be found in Section 4.2 (Experimental Design). Different subscript letters on the graph mean statistically significant difference between the groups (*p* ≤ 0.05).

**Figure 8 ijms-25-06074-f008:**
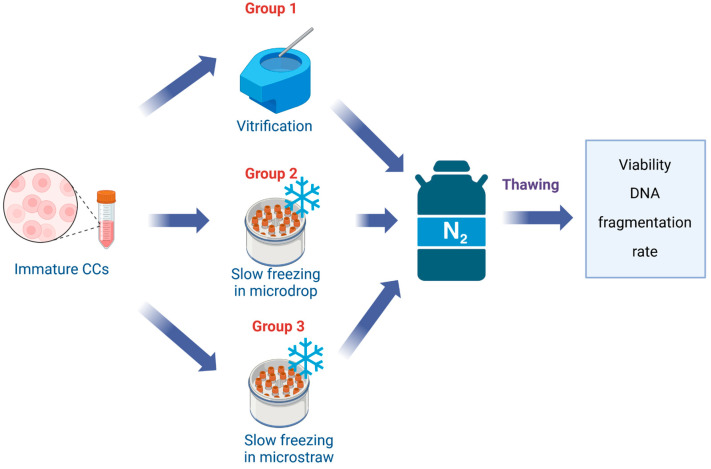
Experimental design for defining the optimal method for cryopreservation of CCs.

**Figure 9 ijms-25-06074-f009:**
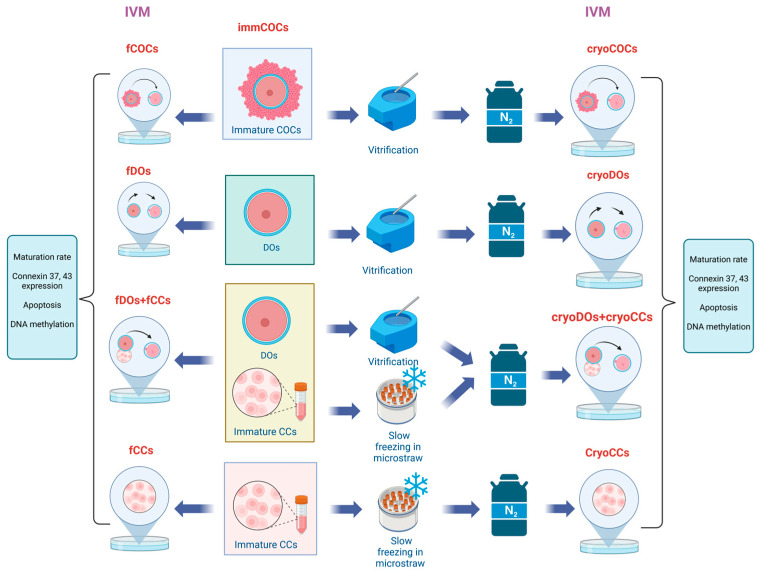
Experimental design for studying the effect of cryopreservation and different IVM systems on the mRNA level of the expression of genes in CCs and oocytes, the localization of connexin, and the oocytes’ maturation rate. Abbreviations used in the scheme: immCOCs, fresh immature COCs; fCOCs, fresh COCs after IVM; cryoCOCs, cryopreserved COCs after IVM; fDOs, freshly denuded oocytes after IVM; cryoDOs, cryopreserved denuded oocytes after IVM; fDOs + fCCs, fresh denuded oocytes matured in vitro with a suspension of fresh CCs; cryoDOs + cryoCCs, cryopreserved oocytes matured in vitro with a suspension of cryopreserved CCs; fCCs, fresh CCs after culturing in IVM media; cryoCCs, cryopreserved cumulus cells after culturing in IVM media.

**Figure 10 ijms-25-06074-f010:**
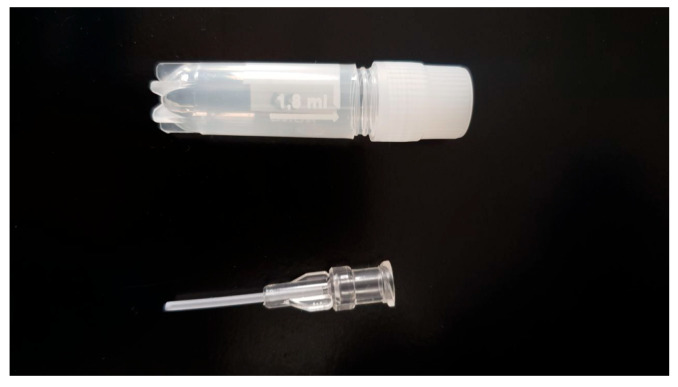
Photo of the microstraw used for slow freezing of CCs in the study.

**Table 1 ijms-25-06074-t001:** Primers used for qRT-PCR of genes in bovine oocytes and cumulus cells.

Gene Symbol	Gene Name	Accession Numer	TaqMan Assay ID	Product Length (bp)
*GJA1 (Cx43)*	Gap junction protein alpha 1	NM_174068.2	Bt03244351_M1	67
*GJA4 (Cx37)*	Gap junction protein alpha 4	NM_001083738.1	Bt03257693_g1	95
*BAX*	BCL2 associated X, apoptosis regulator	NM_173894.1	Bt01016551_g1	72
*BCL2*	BCL2 apoptosis regulator	NM_001166486.1	Bt04298952_M1	72
*DNMT3A*	DNA methyltransferase 3 alpha	NM_001206502.2	Bt01027164_M1	65
*GAPDH*	Glyceraldehyde-3-phosphate dehydrogenase	NM_001034034.2	Bt03210913_g1	66

## Data Availability

The datasets generated during and/or analyzed during the current study are available from the corresponding author upon reasonable request.

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
