# Peer review of "New Approach to the Cryopreservation of GV Oocytes and Cumulus Cells through the Lens of Preserving the Intercellular Gap Junctions Based on the Bovine Model"

_ijms, 2024, doi:10.3390/ijms25116074_

Round 1

Reviewer 1 Report

Comments and Suggestions for Authors

In this ms,  the authors compared different cryopreservation methods and IVM rate of bovine oocytes matured with or without cumulus cell (CC)  supplementation,  either as fresh or cryopreserved cells.They found that the presence of CC during IVM had a beneficial effect on oocyte because they  stimulated Cx37 and 43 mRNA accumulation in oocytes, reduced apoptosis and maintained DNMT3A mRNA level comparable to control.

It is well known from literature data  that in many mammalian species fertilization rate of    DOs  is positively influenced by the presence during maturation and fertilization of  dissociated CCs  that  secrete  factors  improving oocyte nuclear and cytoplasmic maturation. Moreover, in porcine COCs  clustering of Cx43 to lipid rafts occurs in coincidence with resumption of meiosis, followed by their closure about after 18 h of in vitro culture.

Although the  topic of  this paper is of interest, the most prominent limit is that it studied oocytes maturation but not  fertilization in different models of oocyte-cumulus co-culture. In my opinion, it is quite difficult to select  the best experimental protocol for oocyte cryopreservation and IVM without assessing embryonic development.. 

The paper cannot be accepted in the present form, and needs to be accurately revised

Comments:

1.      ABSTRACT:

L14. Remove this sentence because the authors did not demonstrate that coculture   allow gap junction reformation.

L21: what is a” compensatory activity”?  Experiments demonstrating the formation of metabolically-active  gap junctions are lacking.

2.      INTRODUCTION: some relevant literature data are missing. For example:  

a.      Cecconi et al (Endocrinology 2008 Jan;149(1):100-7) showed that ovine CC released paracrine factors able to stimulate oocyte maturation by activating MAPK  

b.      Pereira et al (2019, Rep dom anim 3:53) showed that when equine COCs were vitrified before IVM, CC had a high total DNA fragmentation rate .

c.      López A et al (Porcine Health Manag. 2021 Oct 18;7(1):56) demonstrated that vitrification reduced viability in porcine oocytes and cumulus cells.

d.       In humans, oocytes   in vitro matured with CC obtained from the same COCs  showed changes in gene expression comparable to naturally matured oocytes (Virant Klum RBM 2018 36:508).

e.      Jia et al (MRD 2019, 134:90) analyzed transcriptome  profiles   (RNA-seq) in porcine immature oocytes and in their surrounding cumulus cells (CCs) after vitrification and in vitro maturation (IVM), finding that after IVM  37 genes in MII oocytes and 140 genes in CCs were significantly differentially expressed when  immature COCs  were vitrified at GV.    They included  DNMT3A  in their analysis.

L39. Substitute ref 2 with  recent reference(s). e.g.Pai et al J Hum Reprod Sci 2021,14:30

3.      RESULTS

The number of oocytes utilized in each experiment has not been reported. Please, add this fundamental information. 

Furthermore, Kaiin et al (2020 IOP Conf. Ser.: Earth Environ. Sci. 478 012012) have recently reported a different cryopreservation method for bovine CC that should be cited and discussed. 

a.      Figure 1: in  both A and B the authors used the same black color for  Control (A) and Vitrified  (B) groups. This is confounding.  Tunel test: which cells have been analyzed? Add indications .

b.      Paragraph 2.2 : this paragraph needs to be rewritten. The term “violation of the structure” must be  changed. Morphological alterations could indicate changes to   COC structures. There is no indication in the text of the photos reported in Figure 2, so it is difficult to understand these results.

c.      L 11-113 should be moved to discussion.

d.      L124-131: unclear, please  rewrite . L124: what does it mean “less frequently” referred to protein distribution in DO group?  How many oocytes display this distribution?

e.      Figure 4 A and B and L139. Are you sure that Cx43 expression is higher in CCs? If I compare values reported on vertical axis of the graphs, this statement is not valid.

f.       Figure 5. How values reported in C= BAX/BCL-2 RATIO  have been calculated?

g.      Figure 6B and L179-180: DNMT3A level increases in all somatic cells after IVM,   in COCs as well as in CCs.

4.      DISCUSSION: Since   there are some key questions in Results that need to be answered, my comments to Discussion are limited.

a. L221-228 there are no experiments  in this paper aimed to measure GJ reestablishment.

                b. Under all experimental conditions, oocytes express more Cx37 and Cx43 than CC; Cx37 is  less  expressed than Cx43 in CC and oocytes. This cannot be simply explained by  a  “compensatory   stimulation “.

              M&M:

4.1: Ovaries  and not COCs are transported to labs. Rewrite these sentences.

4.2: classification of experimental groups is difficult and sometimes confused. See for a more clear scheme Zhou et al 2016, Peer J 4:1761, Figure 1.

 L139: “oocyte development”: what do you mean? I imagine that all immature COCs are at GV stage, but no information about this assessment has been provided in the text.

Figure 9: I found this figure quite confused. The term GVs has been used only for DO, but I presume that also COCs are GV arrested.

4.7: how many oocytes have been in vitro matured? How many in each drop ?On which basis the concentration of CC for coculture has been selected? Each bovine COC is surrounded by about 21.000 cells,  (Hashimoto et al Theriogen 1998 49:1451)

I recommended  the authors to add all the information required, and to re-evaluate results . Finally, it is important to highlight that the presence of a polar body  is not indicative of  oocyte quality, which is usually assessed after fertilization  during  embryonic development.

Comments on the Quality of English Language

dear Editof

please find enclosed my comments to the ms. 

Best regards

SC

Author Response

New approach to cryopreservation of GV oocytes and cumulus cells through the prism of preserving the intercellular gap junctions

Response to Reviewer 1 Comments

1. Summary

Thank you very much for taking the time to the comprehensive reviewing of this manuscript. Please find the detailed responses below and the corresponding revisions/corrections highlighted/in track changes in the re-submitted files.

2. Questions for General Evaluation

Reviewer’s Evaluation

Response and Revisions

Does the introduction provide sufficient background and include all relevant references?

Must be improved

Are all the cited references relevant to the research?

Must be improved

Is the research design appropriate?

Yes

Are the methods adequately described?

Must be improved

Are the results clearly presented?

Must be improved

Are the conclusions supported by the results?

Not applicable

3. Point-by-point response to Comments and Suggestions for Authors

Comments 1: L14. Remove this sentence because the authors did not demonstrate that coculture   allow gap junction reformation.

Response 1: Thank you for pointing this out.  In that part of the abstract we assumed that cumulus cells (CCs) and oocytes  “ …co-culture in vitro will contribute to the restoration of lost intercellular gap junction”.  Finally, we have shown that co-culture CCs with oocyte during IVM induced upregulation of RNA expression of gap junction protein genes and localization of proteins was more similar to intact COCs than to denuded oocytes (DOs) without CCs. In our opinion these changes could be considered as a contribution to the restoration of lost contacts.

Comments 2: L21: what is a” compensatory activity”?  Experiments demonstrating the formation of metabolically-active gap junctions are lacking.

Response 2:  We agree with this comment and changed them: “ …. the compensatory upregulation of the genes Cx37 and Cx43 in oocytes to restore gap junctions between cells.”

Comments 3:  INTRODUCTION: some relevant literature data are missing. For example:  

a.      Cecconi et al (Endocrinology 2008 Jan;149(1):100-7) showed that ovine CC released paracrine factors able to stimulate oocyte maturation by activating MAPK 

b.      Pereira et al (2019, Rep dom anim 3:53) showed that when equine COCs were vitrified before IVM, CC had a high total DNA fragmentation rate .

c.      López A et al (Porcine Health Manag. 2021 Oct 18;7(1):56) demonstrated that vitrification reduced viability in porcine oocytes and cumulus cells.

d.       In humans, oocytes   in vitro matured with CC obtained from the same COCs  showed changes in gene expression comparable to naturally matured oocytes (Virant Klum RBM 2018 36:508).

e.      Jia et al (MRD 2019, 134:90) analyzed transcriptome  profiles   (RNA-seq) in porcine immature oocytes and in their surrounding cumulus cells (CCs) after vitrification and in vitro maturation (IVM), finding that after IVM  37 genes in MII oocytes and 140 genes in CCs were significantly differentially expressed when  immature COCs  were vitrified at GV.    They included  DNMT3A  in their analysis.

L39. Substitute ref 2 with  recent reference(s). e.g.Pai et al J Hum Reprod Sci 2021,14:30

Response 3: We are very grateful for these suggestions. As suggested, we have cited these papers in revised version of manuscript. The paper authored by A. López and colleagues (also recommended) was initially mentioned twice in the introduction part.

Comments 4:  RESULTS

The number of oocytes utilized in each experiment has not been reported. Please, add this fundamental information. 

Response 4: We agreed with this reasonable comment and added the oocyte number used for maturation as well as number of replicates of the experiments.

Comments 5:  a. Figure 1: in both A and B the authors used the same black color for Control (A) and Vitrified (B) groups. This is confounding.  Tunel test: which cells have been analyzed? Add indications .

Response 5: Thank you for pointing this out. We have changed the column color in the Figure 1A and 1B, now all of them have the same color. The type of cells was written in the figure title: “The effect of cryopreservation on CCs viability and DNA fragmentation”.  Thus, cumulus cells were assessed by TUNEL test.

Comments 6: Paragraph 2.2 : this paragraph needs to be rewritten. The term “violation of the structure” must be changed. Morphological alterations could indicate changes to   COC structures. There is no indication in the text of the photos reported in Figure 2, so it is difficult to understand these results.

 Response 6: We are very grateful for this comment. We have changed the statement according the Reviewer suggestion: “In the fCOCs group, a large-scale expansion of CCs was observed, while in the cryoCOCs group, morphological alterations of COCs was observed immediately after thawing,….”

We have also added several image numbers to the text: “Figure 2 shows photos of cells before and after IVM. In the fCOCs group, a large-scale expansion of CCs was observed (Figure A, E), …….. their partially detachment from the oocyte and their degradation (Figure 1B), which increased after IVM (Figure 1F). No morphological differences were observed between the fDOs+fCCs and cryoDOs+cryoCCs groups after oocyte maturation (Figure 2G, H). CCs aggregated between themselves and oocytes, forming a structure similar to COCs (Figure 2E).  

Comments 7:   c. L 11-113 should be moved to discussion.

Response 7: We agree and moved this sentence to the discussion part.

Comments 8: d. L124-131: unclear, please rewrite. L124: what does it mean “less frequently” referred to protein distribution in DO group?  How many oocytes display this distribution?

Response 8: We are grateful for this comment and have changed the statement: “The distribution of this protein changed only in the DOs group, where the fluorescence of the Cx37 antibody fluorochrome was slightly detected (Figure 3N,O).” That was observed in all the DOs (n=20).

Comments 9:   e. Figure 4 A and B and L139. Are you sure that Cx43 expression is higher in CCs? If I compare values reported on vertical axis of the graphs, this statement is not valid.

Response 9:  We absolutely agree with this comment and thank you that you pointed out this. We have deleted this sentence.

Comments 10:  f. Figure 5. How values reported in C= BAX/BCL-2 RATIO  have been calculated?

Response 10: We have calculated BAX/BCL-2 RATIO by division of relative RNA expression value of BAX to relative RNA expression value of BCL-2 in each sample.

Comments 11:   g. Figure 6B and L179-180: DNMT3A level increases in all somatic cells after IVM,   in COCs as well as in CCs.

Response 11: Thank you very much for this suggestion we have improved the sentence: “DNMT3a RNA level has increased significantly after IVM in all groups where CCs were used, regardless cryopreservation.”

Comments 12:  4. DISCUSSION: Since   there are some key questions in Results that need to be answered, my comments to Discussion are limited.

a. L221-228 there are no experiments in this paper aimed to measure GJ reestablishment.

Response 12: Thank you very much for this comment. We consider that significant changes in the distribution of Cx 37 and Cx43 as well as in the RNA expression of related genes between DOs and DOs+CCs give us the reason to consider this effect as restoration of GJ.

Comments 13:  b. Under all experimental conditions, oocytes express more Cx37 and Cx43 than CC; Cx37 is  less  expressed than Cx43 in CC and oocytes. This cannot be simply explained by  a  “compensatory   stimulation “.

Response 13: In our study we have compared differences in Cx 37 and Cx 43 expression in fresh and cryopreserved cells. It is well known that Cx 37 expresses more in oocytes than in CCs (Li, T.Y.; Colley, D.; Barr, K.J.; Yee, S.P.; Kidder, G.M. Rescue of oogenesis in Cx37-null mutant mice by oocyte-specific replacement with Cx43. J. Cell Sci. 2007, 120, 4117–4125; Zhao M, Subudeng G, Zhao Y, Hao S, Li H. Effect of Cyclic Adenosine Monophosphate on Connexin 37 Expression in Sheep Cumulus-Oocyte Complexes. J Dev Biol.)  2024;12(2):10. Published 2024 Mar 27. doi:10.3390/jdb12020010 and concentrates mostly between  oocyte and cumulus cells. On the contrary, Cx 43 mostly couples adjacent CCs (Gittens J.E.I., Kidder G.M. Differential contributions of connexin37 and connexin43 to oogenesis revealed in chimeric reaggregated mouse ovaries. J. Cell Sci. 2005;118:5071–5078). That is why expression of Cx43 is supposed to be higher than Cx 37. Our results have also confirmed this. Compensatory upregulated effect on expression of Cx 37 and Cx 43 we have demonstrated in DOs without CCs in culture media. The changes in connexins expression in CCs had different future and can depend on damages caused by cryopreservation or denudation.

Comments 14:  4.1: Ovaries and not COCs are transported to labs. Rewrite these sentences.

Response 14: Thank you very much that you pointed out it. We have corrected this part:” Ovaries were derived from mixed-breed cows and transported from a local slaughterhouse in a warm saline solution (35 °C) to a laboratory COCs were retrieved from follicles ranging in diameter from 2 to 8 mm using an 18 gauge needle attached to a 10 ml syringe.”

Comments 15: 4.2: classification of experimental groups is difficult and sometimes confused. See for a more clear scheme Zhou et al 2016, Peer J 4:1761, Figure 1.

Response 15: Thank you for the comment. Our experimental design is relatively composed, therefor for better understanding we simplify the Figure and add more information to the Figure legend.

Comments 16:  L139: “oocyte development”: what do you mean? I imagine that all immature COCs are at GV stage, but no information about this assessment has been provided in the text.

Response 16: We are grateful for this comment. It seems that you are asking about line 319.

Through gap junctions, CCs can keep the high level of cAMP in the oocyte and hold meiosis at the arrest. Also CCs pass different metabolites that are important for oocyte development and must be accumulated in sufficient quantities prior to further cytoplasmic and nuclear maturation to achieve competence. This we consider as oocyte development.

Comments 17:  Figure 9: I found this figure quite confused. The term GVs has been used only for DO, but I presume that also COCs are GV arrested.

Response 17: Thank you for this comment. All retrieved COCs were randomly divided into the following groups and subjected to the appropriate manipulations. All oocytes were at the same meiosis stage (prophase I, this stage are usually called “germinal vesicle” (GV)).

Comments 18:  4.7: how many oocytes have been in vitro matured? How many in each drop ?On which basis the concentration of CC for coculture has been selected? Each bovine COC is surrounded by about 21.000 cells,  (Hashimoto et al Theriogen 1998 49:1451)

Response 18: Thank you very much that you have driven your attention on these details. We have added the number of oocytes subjected to the IVM in each group.

In fDOs+fCCs group, all CCs left after oocyte denudation were added to the same number oocytes. 10 COCs or 10 DOs were cultured in the 100 µl of IVM media. The size of COCs varied and as we mentioned in the L309-310 : “COCs with at least 3 layers of CCs were selected and used in the experiments. “  But some of COCs were big and had much more than 3 CCs level. That is why it was impossible to obtain the same number of CCs from each COC. We calculated cell the mean concentration of CCs in fDO+fCCs group using hemocytometer (5×105/ml) and added the number of cryopreserved CCs to the oocytes for IVM (CryoDOs+cryoCCs group).

Comments 19:  I recommended  the authors to add all the information required, and to re-evaluate results . Finally, it is important to highlight that the presence of a polar body  is not indicative of  oocyte quality, which is usually assessed after fertilization  during  embryonic development.

Response 19: We absolutely agree that oocyte competence could not be assessed only by the presence of PB. But taking into account that in cryopreserved COCs the maturation rate is extremely low we focused on increasing particular this parameter proposing a new approach for GV oocyte maturation after cryopreservation. Our results have shown that using proposed approach allows improving significantly the maturation rate. Nevertheless, our further research is directed to searching for specific conditions where DOs and CCs can restore connections and structure even more similar to intact COCs to achieve higher results of IVM and then we are going to check fertilization outcomes.

Reviewer 2 Report

Comments and Suggestions for Authors

Review

The authors aimed at determining the optimal method of cryopreservation of the suspension of CCs to mature bovine GV oocytes in vitro and to determine the level of expression of mRNA of genes (GJA1, GJA4; BCL2, BAX) and gene-specific epigenetic marks (DNMT3A) after cryopreservation and IVM in various culture systems. As a result, they found that slow freezing of CCs in microstraws made it possible to preserve the largest number of viable cells and the reconstruction of COCs from cryopreserved CCs and oocytes will provide new opportunities for fertility preservation.

 Although these findings are interesting, the authors should carefully consider the following points for revision.

Major Comment

In this study. authors evaluated oocytes and CCs after IVM based on gene expression levels of GJA1, GJA4, BCL2, BAX, and DNMT3A. However, it should be noted that the expression of these genes does not guarantee fertilization and embryonic development potentials.

If IVF data were provided, this method could be more convincing.

Minor comments

1.    The title or abstract should include the species like “cow” or “bovine”.

2.    P1 Line 41, “since are” seems not perfect sense.

3.    P2 Lines 79-86, please more kindly describe the results. It is difficult to understand what the authors did and how they analyzed since they started to mention just the “viability”. Even if the details are written in the Materials and Methods, they should briefly mention about experiment methods.

4.    P3, Figure 1, what are the positive and negative control? Also “Chanel GFP” means. The authors used GFP for this assay?  

5.    P3 Lines 107-108, “after IVM … and after cryopreservation” may need to be fixed to make sense.   

6.    P3 Lines 110 and 111, After “maturation” and “COCs”, please cite the figure numbers.

7.    P3 Figure 2, rather than the current “K” and “L”, photos for the denuded oocytes should be shown.

8.    P3 Figure 2, what does “after heating” mean?

9.    P4 Lines 118-119, although “The expression level of the Cx 37 mRNA was significantly higher in CCs after maturation in all study groups compared to immature CCs”. fCOC is not significantly high as Figure 2b.

10.  P4 Figure 4B some of “CryoDOs+cryoCCs” is hided.

11.  P5 Line 139, “In CCs, the expression of Cx 43 was higher, compared to oocytes” but apparently Figure 4 A and B does not support this

12.  P8 line 240, “led” may be “lead”.

13.  P10 Fig 8 if possible, “Slow freezing in microdrop” and “Slow freezing in microstraw” should be differently illustrated.

14.  P11 Fig 9 “Slow freezing” should be described either by “microdroplets” or “microstraws”.

15.  P10 Section “4 Materials and Methods” should be moved before results as section “2”. Then, “3 Discussion” and “5 Conclusions” should be connected directly with each other.

Author Response

mulus cells through the prism of preserving the intercellular gap junctions

Response to Reviewer 2 Comments

1. Summary

Thank you very much for taking the time to the comprehensive reviewing of this manuscript. Please find the detailed responses below and the corresponding revisions/corrections highlighted/in track changes in the re-submitted files.

2. Questions for General Evaluation

Reviewer’s Evaluation

Response and Revisions

Does the introduction provide sufficient background and include all relevant references?

Yes

Are all the cited references relevant to the research?

Yes

Is the research design appropriate?

Can be improved

Are the methods adequately described?

Yes

Are the results clearly presented?

Can be improved

Are the conclusions supported by the results?

Can be improved

3. Point-by-point response to Comments and Suggestions for Authors

Comments 1: Major Comment

In this study authors evaluated oocytes and CCs after IVM based on gene expression levels of GJA1, GJA4, BCL2, BAX, and DNMT3A. However, it should be noted that the expression of these genes does not guarantee fertilization and embryonic development potentials.

If IVF data were provided, this method could be more convincing.

Response 1: We are grateful for this reasonable comment and agree but taking into account that cryopreservation of COCs leads to detrimental decreasing of oocyte maturation we focused mainly on improvement of this parameter. We cannot expect to receive high fertilization and embryo development rate without high maturation value. Despite our new approach allowed improving it, still it is lower than in control (fresh) cells. Our future study is devoted to find such a system where oocytes could restore better connections with CCs cryopreserved by proposed here method. We are going to check fertilization outcomes after we have reached the maturation rate comparable to the control.

Comments 2: The title or abstract should include the species like “cow” or “bovine”.

Response 2:  Thank you very much for this suggestion. Indeed, in our study we used only bovine oocytes but taking into account that we used this species as a model object for cryopreservation of human immature oocytes and IVM we can propose such kind of title: New approach to cryopreservation of GV oocytes and cumulus cells through the prism of preserving the intercellular gap junctions based on the bovine model

If you have any other suggestion we will be happy to follow it.

Comments 3:  P1 Line 41, “since are” seems not perfect sense.

Response 3: Thank you very much that you have pointed out on it.

We have corrected:

“In the case of tissues, the effectiveness of the freezing method can be lower, since they are composed of different types of cells….”

Comments 4:  P2 Lines 79-86, please more kindly describe the results. It is difficult to understand what the authors did and how they analyzed since they started to mention just the “viability”. Even if the details are written in the Materials and Methods, they should briefly mention about experiment methods.

Response 4: Thank you for this advice, this paragraph has been corrected: “At the first stage of our study we compared the efficiency of using different methods (vitrification, slow freezing in microdrops and slow freezing in microstraws) for CCs preservation. The viability was measured in CCs after warming by Trypan blue staining. As shown in Fig 1A the highest value of this parameter was obtained after slow freezing in microstraw.

Comments 5:  P3, Figure 1, what are the positive and negative control? Also “Chanel GFP” means. The authors used GFP for this assay? 

Response 5:  We are grateful for the comment and added the description of the positive and negative control to the M&M: “The positive and negative controls were incubated with DNase I (deoxyribonuclease I, 100 IU/mL; Invitrogen, Carlsbad, CA, USA) at 37 ℃ for 1 h. Then after washing in PBST cells from positive control were incubated with TUNEL reaction mixture containing FITC-conjugated dUTP (label solution) and terminal deoxynucleotidyl transferase at 37 ℃ for 1 h in the dark. The negative control was incubated only with label solution in the absence of terminal deoxynucleotidyl transferase. All samples were stained with 10 mg/ml Hoechst (33342, Sigma–Aldrich, USA) in PBS during 15 min at room temperature. The excitation 488 nm wavelength was used to identify TUNEL staining and the 352 nm wavelength was used to distinguish nuclei (Figure 1).”

We have not used GFP staining but used the same wavelength for the excitation. We have corrected the Chanel name on the Figure 1.

Comments 6:  P3 Lines 107-108, “after IVM … and after cryopreservation” may need to be fixed to make sense.   

Response 6: Thank you very much for this comment we have change the figure description: “Figure 2. Morphology of oocytes and CCs of cows cryopreserved and cultured in various in vitro systems. A ‒ immCOCs immediately after retrieval; B ‒ immCOCs immediately after warming before IVM; C ‒ fresh immDOs with CCs from immCOCs before IVM; D ‒ cryopreserved immDOs with CCs from immCOCs before IVM; E ‒ fCOCs after IVM; F ‒ cryoCOCs after IVM; G ‒ fDOs with fCCs after IVM; H ‒ CryoDOs with cryoCCs after IVM; I ‒ fDOs after IVM; J ‒ cryoDOs after IVM; K ‒ fCCs cultured 24h in IVM medium; L ‒cryoCCs cultured 24hin IVM medium. The group description could be found in the 4.2 Experimental design subsection.” 

Comments 7: P3 Lines 110 and 111, After “maturation” and “COCs”, please cite the figure numbers.

 Response 7: We agree we this comment and made appropriate changes in the text : “No morphological differences were observed between the fDOs+fCCs and cryoDOs+cryoCCs groups after oocyte maturation (Figure 2G, H). CCs aggregated be-tween themselves and oocytes, forming a structure similar to COCs (Figure 2E).”

Comments 8:   P3 Figure 2, rather than the current “K” and “L”, photos for the denuded oocytes should be shown.

Response 8: We cultures either fresh or cryopreserved CCs without oocyte in the IVM media to see the oocyte effect on CCs ability to form aggregates. Thus, photos “K” and “L” on Figure 2 represent only CCs.

Comments 9: P3 Figure 2, what does “after heating” mean?

Response 9: Thank you very much that you pointed out it, we have corrected as you can see in Response 6.

Comments 10:  P4 Lines 118-119, although “The expression level of the Cx 37 mRNA was significantly higher in CCs after maturation in all study groups compared to immature CCs”. fCOC is not significantly high as Figure 2b.

Response 10:  We are grateful for this comment. Yes, you are absolutely right, we have corrected the sentence.: “The expression level of the Cx 37 mRNA was significantly higher in CCs after maturation in all study groups, except fCOCs, compared to immature CCs”

Comments 11:  P4 Figure 4B some of “CryoDOs+cryoCCs” is hided.

Response 11: It seems that there is some technical issue after transforming .docx file into pdf and the column was not visible. We hope this time after revision it will be correct.

Comments 12:  P5 Line 139, “In CCs, the expression of Cx 43 was higher, compared to oocytes” but apparently Figure 4 A and B does not support this

Response 12: We absolutely agree with you and removed the statement.

Comments 13:  P8 line 240, “led” may be “lead”.

Response 13: Thank you very much. We totally agree and corrected the sentence.

Comments 14:  P10 Fig 8 if possible, “Slow freezing in microdrop” and “Slow freezing in microstraw” should be differently illustrated.

Response 14: Thank you for this suggestion. We initially tried this, but due to the small size of the drop (10 µl) and the semi-transparency of the cryovial, the drop was barely visible. We have included a photo of a microstraw to demonstrate the system we used as the microstraw is not produced for cryopreservation purposes.

Comments 15: P11 Fig 9 “Slow freezing” should be described either by “microdroplets” or “microstraws”.

Response 15: Thank you for the comment we have added this to the scheme.

Comments 16: P10 Section “4 Materials and Methods” should be moved before results as section “2”. Then, “3 Discussion” and “5 Conclusions” should be connected directly with each other.

Response 16: Thank you very much for this suggestion. We also think that it would be better to describe first “Materials and Methods” and then the “Results” and “Discussion”. The IJMS provides a template for a paper format and structure that we followed, it was not our initiative.

Round 2

Reviewer 1 Report

Comments and Suggestions for Authors

The authors have addressed most of the comments. I think the paper can be accepted for publication.